

# Development of Super-Ensemble techniques for ocean analyses: the Mediterranean Sea case

Jenny Pistoia[1], Nadia Pinardi [2,3], Paolo Oddo [1,*], Matthew Collins[4], Gerasimos Korres[5], and Yann Drillet[6]

[1]Istituto Nazionale Geofisica e Vulcanologia, Bologna, Italy
[2]Department of Physics and Astronomy, University of Bologna,Italy
[3]Centro euro-Mediterraneo sui Cambiamenti Climatici, Via Augusto Imperatore, 16 - 73100 Lecce,Italy
[4]College of Engineering, Mathematics and Physical Sciences, University of Exeter, Exeter,UK
[5]Hellenic Center for Marine Research, 46,7 Km Athens-Sounion Road, Anavissos, 19013, Greece
[6]Mercator Ocean Parc Technologique du Canal 8-10 rue Hermes 31520 Ramonville St Agne,France
[*]Now at : NATO Science and Technology Organization Centre for Maritime Research and Experimentation, La Spezia, Italy

*Correspondence to:* Jenny Pistoia Istituto Nazionale Geofisica e Vulcanologia, Bologna, Italy(E-mail: jenny.pistoia@ingv.it)

**Abstract.** A super-ensemble methodology is proposed to improve the quality of short-term ocean analyses for Sea Surface Temperature (SST) in the Mediterranean Sea. The methodology consists in a multi-linear regression technique applied to a Multi-Physics Multi-Model Super-Ensemble (MMSE) dataset. This is a collection of different operational forecasting analyses together with ad-hoc simulations, created by modifying selected numerical model parameterizations. A new linear regression

algorithm based on Empirical Orthogonal Function filtering techniques is shown to be efficient in preventing overfitting problems, although the best performance is achieved when a simple spatial filter is applied after the linear regression.Our results show that the MMSE methodology improves the ocean analysis SST estimates with respect to the Best Ensemble Member (BEM) and that the performance is dependent on the selection of an unbiased operator and the length of training. The quality of the MMSE dataset has the largest impact on the MMSE analysis Root Mean Square Error (RMSE) evaluated with respect

to observed satellite SST. The MMSE analysis estimates are also affected by changing the training period with the longest period leading to the smoothest estimates. Finally, lower RMSE analysis estimates result from the following: a 15 days training period, an overconfident MMSE dataset (a subset with the higher quality ensemble members), and the least square algorithm being filtered a posteriori.

**Keywords.** Ocean analysis skill - Superensemble - Dataset composition - Overfitting

**1   Introduction**

The limiting factors to short-term ocean forecasting predictability are the uncertainties in ocean initial conditions, atmospheric forcing Pinardi et al. (2011), lateral boundary conditions tighter with numerical model representation and numerical inaccuracies. To assess and control these uncertainties, an ensemble approach can be used as shown, for example, by Kalnay and Ham (1989) where the simple ensemble mean is shown to have a smaller Root Mean Square Error (RMSE) than each contribut-

ing member. The assumption that different models may have complementary forecasting and analysis skills emerged from the pioneering work of Lorenz (1963), when the notion of an ensemble forecast was first described, which was obtained by



factorizing all the members' performances. Most common ensemble forecasts came from a single model running with a set of perturbed initial, lateral or vertical boundary conditions. Hence the implicit hypothesis is that forecast errors arise from inaccurate initial/boundary conditions, while the model is considered as being perfect. Accounting for the model error was the first step in multi-model ensemble forecasting. Feddersen et al. (1999) reported that the low ensemble spread is likely to be

produced by correlated models, hence only a set of different models is expected to reduce the model systematic error. Shukla et al. (2000) proposed a combination of member predictions with similar forecast skills in order to further reduce the posterior forecast error calculated from the multi-model ensemble. The optimal combination of several model outputs (each with its own strengths and weaknesses) that can sample the forecast uncertainty space is the underlying idea behind Super Ensemble (SE) estimates. In this paper we used the heuristic SE concept of Krishnamurti et al. (1999) where different and independent model

forecast members are merged using a multiple linear regression algorithm. This method led to a reduced forecast RMSE for the 850-hPa meridional wind and hurricane tracking in the Northern Hemisphere. The Multi-Model Super Ensemble (MMSE) approach was then subsequently (Evans et al., 2000; Krishnamurti et al., 2000; Stensrud et al., 2000). While ensemble techniques are routinely used in operational weather forecasting (Toth and Kalnay, 1997; Stephenson and Doblas-Reyes, 2000), SE and MMSE approaches have mostly been applied in seasonal studies (Krishnamurti et al., 1999; Stefanova and Krishnamurti,

2002; Pavan and Doblas-Reyes, 2000; Kharin and Zwiers, 2002) . Only preliminary work has been carried out in ocean forecasting. The work of Rixen et al. (2009) is a reference for temperature predictions, simulated ocean trajectory SE methods are described in Vandenbulcke et al. (2009), while Lenartz et al. (2010) presents a SE technique with a Kalman filter to adjust three dimensional model weights. Rixen et al. (2008) introduced the concept of a hyper-ensemble, which combines atmospheric and oceanographic model outputs. In this paper we develop a new MMSE method to estimate Sea Surface Temperature (SST) as

this is an important product of ocean analysis systems with multiple users. Accurate knowledge of SST is fundamental both for climate and meteorological forecasting, therefore increasing the capacity of SST analyses is crucial for the uptake of operational products. A MMSE dataset is constructed to sample the major error sources for the SST forecast and a new linear regression algorithm is developed and calibrated. After a description of the multi-model data set in Section 2, a comprehensive explanation of the super-ensemble technique is reported in Section 3. Sensitivity studies on SE algorithm choices are proposed

in Section 4. Finally, conclusions are drawn in Section 5

## 2 Multi-Model Multi-Physics Dataset

The MMSE dataset includes the collection of daily mean outputs from five operational analysis systems in the Mediterranean Sea and four outputs from the same operational forecasting model but with different physical parameterization choices. The study period lasts from 1 January 2008 to 31 December 2008. The main differences between the MMSE members are mainly

due to the different numerical schemes used, the data assimilation scheme and the model physical parameterizations. Optimally Interpolated satellite SST observations (OI-SST) (Marullo et al., 2007) are used as the truth estimator and the model outputs are compared with the satellite OI-SST to assess their quality. The main characteristics of MMSE members are listed in Table 1, while a more detailed description of the originating analysis and forecasting systems can be found in Appendix A. Our aim




is to estimate the most accurate daily SST for a 10-day analysis period that took place after a training period defined in the past. The resulting MMSE estimate is also called a posterior analysis. The similarities and differences between MMSE members and the OI-SST data set is quantified in terms of the anomaly correlation coefficient (ACC), root-mean-square error (RMSE) and normalized standard deviation (STD). These statistical scores are listed in Table 2 for each MMSE member for the whole

of 2008, with the seasonal cycle removed. MercatorV0 best reproduces the STD of the observations, while INGV-SYS4a3 analysis has a higher ACC and lower RMSE. Thus hereafter INGV-SYS4a3 will be called the Best Ensemble Member (BEM).

## 3   Super Ensemble Methodology

Our SE methodology is based on Krishnamurti et al. (1999). Let us call $S1_t$ the SE estimate of a model state variable, $F_{i,t}$ the model state at time 't' and for the 'i-th' model. Let us define two different periods, the training and analysis periods, the former

period preceding the target analysis period. A $S1_t$ estimator is then defined as:

$$S1_t = \overline{O} + \sum_{i=1}^{N} a_i \left( F_{i,t} - \overline{F_i} \right)$$
$$\overline{O} = \frac{1}{M} \sum_{t=1}^{M} O_t \tag{1}$$

where $\overline{F_i}$ and $\overline{O}$ are the time mean over the training period, as defined in Appendix B, $a_i$ are the regression coefficients, $N$ is the number of SE members and $M$ is the number of training period days. The regression is unbiased because the time mean of the dataset is removed and only model field anomalies are used. The regression coefficients are computed as a classical

multilinear ordinary least squares problem. Let us define the covariances of the model ensemble members as:

$$\sum_{t=1}^{M} \left( F_{i,t} - \overline{F_i} \right) \left( F_{j,t} - \overline{F_j} \right) \equiv \Gamma_{i,j} \tag{2}$$

and the covariance between observations and model anomalies

$$\sum_{t=1}^{M} \left( F_{j,t} - \overline{F_j} \right) \left( O_t - \overline{O} \right) \equiv \phi_j \tag{3}$$

The regression coefficients are then written as:

$$a_i = \left( \Gamma_{i,j} \right)^{-1} \cdot \phi_j = \sum_{t=1}^{M} \left( \left( F_{i,t} - \overline{F_i} \right) \left( F_{j,t} - \overline{F_j} \right) \right)^{-1} \left( \left( F_{j,t} - \overline{F_j} \right) \left( O_t - \overline{O} \right) \right) \tag{4}$$

Yun et al. (2003) reported a skill improvement in the SE algorithm when the seasonal signal is removed prior to the regression procedure. The second SE method, called $S2_t$, uses the same regression algorithm of Eq.4, but with the seasonal cycle subtracted instead of the training period time mean. The definition of the new unbiased estimator is presented in Appendix B. Kharin and Zwiers (2002) suggested that the poor performance of MMSE algorithms is due to overfitting i.e. biased estimates

of the regression coefficients. This means that not all the model members and the observations used in the estimation are really independent and the matrix inversion in Eq. 4 is near to being singular. In order to reduce overfitting, several methods have been proposed. Following Boria et al. (2014) who used a spatial filter to reduce overfitting in ecological niche models, we



developed a new method, called S3, which filters the S2 estimates with a simple spatial median filter with a radius of 12km. This radius is related to the first baroclinic Rossby radius of deformation in the Mediterranean Sea (e.g., Robinson et al., 1987; Pinardi and Masetti, 2000). A principal component regression coefficient technique defined by von Storch and Navarra (1995) offers an alternative way of performing the regression ensuring more uncorrelated variables. In our formulation it is suggested that the $F_{i,t}$ are decomposed into horizontal Empirical Orthogonal Function (EOF) mode singular vectors of a data matrix which contains the training period model and observed fields. Thus we form a state variable vector $\Theta$ that contains model and observation anomalies for the training period:

$$\Theta = \left[ O'_t F'_{i,t} \right] \tag{5}$$

Here the variables $O'_t F'_{i,t}$ indicate anomalies with respect to the seasonal and the training period. Decomposing the matrix $\Theta$ into a horizontal EOF, called $eof(x,y)$, and temporal amplitudes, $\Theta$ we can write the least square solution of Eq. 4 computed for the amplitudes of the spatial EOFs. The $O'_t$ and $F'_{i,t}$ fields will be projected into the retained $eof(x,y)$ to obtain:

$$F'_{i,m} = \sum_{k=1}^{p} \alpha_{k,m}(t_i)\, eof_k(x,y); \qquad O'_i = \sum_{k=1}^{p} \beta_k(t_i)\, eof_k(x,y) \tag{6}$$

$$\sum_{t=1}^{M} \alpha_{j,t}\alpha_{j,t} = \Gamma_{i,j}^{eof} \tag{7}$$

$$\sum_{t=1}^{M} \alpha_{i,t}\beta_{j,t} = \phi_{j}^{eof} \tag{8}$$

The regression coefficients are now written for each $eof$ component as follows:

$$a_i^{eof} = \left( \Gamma_{i,j}^{eof} \right)^{-1} \cdot \phi_j^{eof} = \left( \sum_{t=1}^{M} \alpha_{i,t}\alpha_{j,t} \right)^{-1} \sum_{t=1}^{M} \alpha_{j,t}\beta_{j,t} \tag{9}$$

A new SE estimate, S4, is now defined as:

$$S4_t = \overline{O'} + \sum_{i=1}^{N} a_i^{eof} \cdot \left( F'_{i,t} - \overline{F}'_i \right) \tag{10}$$

The different statistical regression algorithms are summarized in Table 3.

## 4   MMSE experiments

In this section we describe the MMSE experiments carried out to test the four regression algorithms. For all our regression algorithms, we selected a test analysis period from 25 April to 4 May 2008, while the related training period was chosen as a number n of days before April 25, depending on the different experiments in order to test sensitivity on the selected training period length. Our results refer to the whole of 2008, where ten days analyses were performed twice a week after the computation of weights during the training period.


### 4.1 Classical SE method experiments

On the basis of a 15-day training period, Figure 1 shows the S1 and S2 posterior estimates for the first day of the test analysis period. S1 and S2 reconstructed SST fields are very noisy compared to observations and both S1 and S2 are clearly worse than the BEM estimate. The two estimates at the end of the test analysis period are shown Figure 2, where the overfitting problem

is even more evident. The field noise can be reduced by lengthening the training period to 35 days, as shown in Figure 3 and in Figure 4. Both S1 and S2 predictions show a reduction in the warm bias with respect to the BEM in the eastern Mediterranean (Figure 1 and Figure 2). However S1 does not show any improvement in terms of RMSE and ACC (data not shown). In order to examine the effect of the specific MMSE members on the S1 estimation performance, we create three different MMSE datasets (Table 4). Dataset A corresponds to an overconfident (Weigel et al., 2008) dataset or correlated ensemble members.

We consider dataset B, which is well-dispersed, to be the best and the worst ensemble member, together with other correlated members i.e. with similar RMSE standard deviation (STD) and ACC (see Table 4), while dataset C, which is poorly-dispersed, considers worst members together with correlated members.

To quantify the differences between the three data sets, a bias indicator $d$ is estimated. This value corresponds to the domain averaged SST difference as follows:

$$d = \langle \frac{F_t(x,y) - O_t(x,y)}{O_t(x,y)} \rangle \tag{11}$$

where the triangular parentheses indicate the domain average. If $d$ is close to 0, this means a small data set bias, while a positive (negative) d means a SST positive (negative) bias. Figure 5 shows the distributions of d for different MMSE datasets. The distributions are bimodal with two maxima, the first around $0^oC$ and the second around $0.05^oC$, which means that the dataset statistically tends to overestimate the spatial mean SST. From these distributions we can also observe that:

– dataset A has the smallest bias because the d=$0^oC$ peak is larger than the peak at $0.05^oC$;

   – for dataset B, which is constructed from well-dispersed model ensemble members, the two peaks become of equivalent amplitude and

   – for dataset C, which is constructed from badly-dispersed model ensemble members, the estimate enhances the positive bias.

Figure 6 presents the histogram of d for the S1 estimates on the first day of analysis for the whole of 2008 as a function of the full and subsampled MMSE datasets. It is evident that all the MMSE datasets give a S1 distribution peak around d=0 $^oC$ with a strong reduction in the distribution width. This means that the algorithm is capable of neglecting the information from biased members. The small S1 bias remains the same till the fifth day of the analysis period (not shown) after which the performance of the algorithm starts to deteriorate. On the last day, there is a nearly flat histogram (Figure 7 ). This means that unbiased

analyses can be produced for up to five days with a 15 days training period, no matter which MMSE data set is used. With respect to RMSE, different MMSE datasets and training period lengths give different results already on day 1 of the analysis period, as shown in Table 5, both for S1 and S2 posterior analysis estimates. It is now evident that the overconfident dataset and





the longest training period (35 days) produce on average the lowest RMSE values during the analysis period. In conclusion, the Krishnamurti et al. (1999) method can be applied relatively successfully to the oceanic multi-model state estimation case, using at least a 14 (and up to 35) day training period and with only a 5-member ensemble dataset if the quality of the chosen members is high. However, the S1 and S2 estimates are both affected by noise and only a modification in the regression method
will lead to a low RMSE posterior analysis noiseless estimate.

## 4.2   New SE method experiments

In order to reduce the overfitting of the SE estimate, here we show the results of the S3 and S4 algorithms. Both proposed methodologies are used with the overconfident dataset (dataset A) and a 15-day training period. In S4, the number of retained EOFs is changed for each experiment and this number is chosen in order to account for 99.5% of the system variance. Figure
8 shows the number of retained EOFs as a function of seasons and for different training period lengths. The minimum number of retained EOFs is 46 with a 15-day training period, while the maximum number is 164 obtained with 35 days. As expected, the number of EOFs retained increases when we extend the training period, with some variability during the year. Usually the minimum number of EOFs was found in the summer. The S3 and S4 posterior estimates are shown in Figure 9 and Figure 10 for the first day of the test analysis period. Both estimates are much smoother than the equivalent S1 and S2 estimates in
Figure 1. The S3 also seems to be less biased with respect to observations. A map of the differences between the truth and the SE estimates highlights the better performances of S3 compared to S4 for the whole test analysis period (not shown). Error statistics for the various methods were computed for all of 2008, we again produced a 10-day analysis from the overconfident training dataset A every 4 days with a variable training period from 15 to 35 days. The RMSE is shown Figure 11. The best SE method is given by S3 which has about half of the RMSE value of the BEM for the whole of 2008. This is due to the fact that
filtering acts as a smoother by keeping the large-scale bias small, while EOFs do not control bias at the large scale. Following Murphy (1993) we evaluate the ACC in order to assess the "consistency" of the proposed SuperEnsemble estimate. ACC values are listed in Table 6. As a result of this skill, S3 decreased its consistency in relation to an increase of the length of the training period. On the other hand, S4 had a constant ACC irrespectively of the chosen training period. However B.E.M was the most "consistent" member. This means that although our SE can be used as a statistical tool, physical constraints are needed in order
to have more consistent maps too. Nevertheless it should be highlighted that S1 and S2 are even less consistent compared to the worst contributing member. BIAS skills were also evaluated for the proposed methodologies, however as expected due to their construction, all the SE estimates were unbiased. Thus no inference can be drawn from the BIAS skills.

## 5   Conclusions

We developed a Multi-Model Multi-Physics Super-Ensemble (MMSE) methodology to estimate the best SST from different
oceanic analysis systems. Several regression algorithms were analyzed for a test period and the whole of 2008. We examined different conditions when the MMSE estimate outperforms the BEM of the generating ensemble. The target was to obtain 10-day posterior analyses using a training period in the past for the regression algorithm and to generate the lowest bias and



RMSE for the MMSE estimates. The results show that the ensemble size, quality and type of members, and the training period length are all-important elements of the MMSE methodology and require careful calibration. Almost 2000 posterior analyses were produced for 2008 with different training periods. The classical SE approach, as proposed by Krishnamurti et al. (1999) , here called the S1 estimate, cannot be successfully applied to estimate oceanic MMSE SST. An initial improvement to S1,

named S2, is the subtraction of the seasonal signal in the ensemble members and the unbiased estimator. This leads to a strong reduction in RMSE (more than 20%) but the resulting field is noisy compared to observations. This is the well-known overfitting problem of the technique described in Kharin and Zwiers (2002). The further modification of S2 using a simple spatial filter, names S3, can give lower RMSE values than the BEM for the entire 10-day analysis period. A new methodology, based on EOFs, named S4, also reduces the RMSE. However S3 outperforms S4 and could represent a practical technique for

applications in operational oceanographic analyses for up to 10 days on the basis of the previous 15 days of analyses. This is only a starting point. MMSE techniques for ocean state estimation problems require further study before optimal methods can be found. In this paper we show that with a rather limited but overconfident dataset (with a low bias of the starting ensemble members), the RMSE analysis can be improved. This posterior value added estimation could, for example, be used to produce a more accurate MMSE analysis dataset. Future developments could involve the addition of physical constraints during the

regression, considering for example cross-correlations with atmospheric forcing. MMSE should also be applied to the ocean forecast problem instead of the analysis problem. The difference for MMSE forecast estimates is that atmospheric forecast uncertainties are not contained in training period analyses, and the size of the ensemble members required could increase considerably, as well as the complexity of the estimation problem.

**Data availability**

The full dataset can be found at http://oceanop.bo.ingv.it/mmse-dataset/ . Algorithms available upon request.

*Author contributions.* Jenny Pistoia with the supervision of Dr. Paolo Oddo and Professor Nadia Pinardi designed and performed all the INGV-Multi Physisc simulations and collected all the INGV analysis. Professor Matthew Collins supervised all the activity connected with the SE based on EOFs. Gerasimos Korres and Yann Drillet provide respectively the HCMR and Mercator analysis members in the same grid of INGV members in order to build the MMSE dataset. Jenny Pistoia prepared the manuscript with contributions from all co-authors.

*Acknowledgements.* This work was supported by the University of Bologna as part of the graduate program in geophysics and by the MyOcean2 Project. The CMCC Gemina project funded Dr. Pistoia's studies at the University of Exeter. Publication was supported from the Italian Ministry of Education, University and Research under the project RITMARE.



## Appendix A: Dataset description

The analysis systems that generated the ensemble members of the experiments used in this paper are briefly described below:

- SYS3a2: System composed of the numerical code of OPA8.2 implemented in the Mediterranean Sea (Tonani et al., 2008) and 3DVAR assimilation scheme (Dobricic and Pinardi, 2008);

- SYS4a3: uses NEMO 2.3 (Oddo et al., 2009) as a numerical model and also 3DVAR for the assimilation. Other differences are due to the different boundary conditions, since here the model is nested within the monthly mean climatological field computed from the daily output of the MERCATOR $\frac{1}{4}^o$ resolution global model;

- Mercator V0 (PSY2V3R1): The numerical code is based on NEMO 1.09 and it is implemented in the North Atlantic and Mediterranean Sea with a horizontal resolution of $\frac{1}{12}^o$ and 50 vertical levels. Real time system from 2008 to 2010, was initialized for a calibration phase in 2006. ECMWF analysis and forecasts are coupled daily. The data assimilation scheme is based on a local Singular Evolutive Extended Kalman Filter (Pham et al., 1998) and the observation assimilated are in situ Temperature and Salinity profiles in the CORIOLIS database, together with SST and along track SLA. Further details are described in Brasseur et al. (2005) .

- Mercator V1(PSY2V4R1): The numerical code is based on NEMO 3.1 version and it is implemented in the North Atlantic and Mediterranean Sea with a horizontal resolution of $\frac{1}{12}^o$ and 50 vertical levels. The real time system was initialized in October 2006 from a 3D climatology of temperature and salinity (Levitus et al., 2005) providind analysis and forecast from 2010 to 2013. The code includes several improvement in the model configuration as open boundary condition (from the global system) and higher frequency atmospheric forcing (each 3hours). The data assimilation scheme it's pretty closer to the previous version plus a bias correction based on 3Dvar assimilation scheme (Lellouche et al., 2013)

- HCMR: Hellenic Centre for Marine Research (Korres et al., 2009). The Mediterranean Sea model is based on Princeton Ocean Model (POM) code a primitive equations 3-D model using Mellor-Yamada 2.5 turbulence closure scheme. The model has a bottom - following vertical sigma coordinate system, a free surface and a split mode time step. Potential temperature, salinity, velocity and surface elevation, are prognostic variables. The model has a horizontal resolution of $\frac{1}{10}^o$ and 25 sigma layers along the vertical with a logarithmic distribution near the surface and the bottom. The model includes parameterization of the main Mediterranean rivers while the inflow/outflow at the Dardanelles is treated with open boundary techniques. The Mediterranean model is forced with hourly surface fluxes of momentum, heat and water provided by the Poseidon - ETA high resolution ($\frac{1}{20}^o$) regional atmospheric model (A. and P., 2009) issuing forecasts for 5 days ahead. The assimilation system for the Mediterranean Sea hydrodynamic model is very similar to the one presented in the work of Korres et al. (2010). It is based on the SEEK filter with covariance localization and partial evolution of the correction directions. The error covariance matrix is approximated with 60 EOF modes (correction directions) where the first 18 (the most dominant ones) are evolved with the model dynamics while the rest are kept invariant in time. The localization technique adopted for the Mediterranean Sea forecasting system is explained in (Korres et al., 2010). The method localizes the covariance matrix by neglecting observations beyond a cut-off radius which is selected upon sensitivity studies to be equal to 200 km.

- NEMO Multi-Physics: This is the same as SYS4a3 NEMO 2.3 code without assimilation but with different model physical parameterizations.



## Appendix B:  Algorithm time averages and projection on EOFs

Here we show how the observed and model fields are decomposed into different temporal signals. Let us consider to be the daily OI-SST and one of the model members daily mean SST. We decompose the signal into seasonal mean, training mean and anomaly. Considering the two time average operators:

5  $$\langle f(x,y) \rangle_s = \frac{1}{q} \sum_{t=1}^{q} f(x,y,t)) \tag{B1}$$

$$\langle f(x,y) \rangle_{TR} = \frac{1}{N} \sum_{t=1}^{N} f(x,y,t)) \tag{B2}$$

where "q" is the number of days in the month of the year with a value evaluated over a long time-series mean from 2001 to 2007. and N is the number of training days, we write:

$$O(x,y,t) = \langle O(x,y) \rangle_S + \langle O(x,y) \rangle_{TR} + O'(x,y,t)$$
$$F(x,y,t) = \langle F(x,y) \rangle_S + \langle F(x,y) \rangle_{TR} + F'(x,y,t) \tag{B3}$$

10    The last term on the right of B3 is the anomaly term used in Eq. 4 and 9





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




| MMSE member name | CODE | Vertical mixing scheme | Horizontal Diffusion | Horizontal Viscosity | Assimilation scheme |
|---|---|---|---|---|---|
| INGV-SYS3a2 | OPA 8.2 (Madec et al., 1998) | (Pacanowski and Philander, 1981) | Bilaplacian | Bilaplacian | 3Dvar (Dobricic and Pinardi, 2008) |
| INGV-SYS4a3 | NEMO 2.3 (Madec, 2008) | (Pacanowski and Philander, 1981) | Bilaplacian | Bilaplacian | 3Dvar (Dobricic and Pinardi, 2008) |
| Mercator-V0 | NEMO 1.09 (Madec, 2008) | k-esspilon (Gaspar et al., 1990) | Bilaplacian | Isopycnal laplacian | SAM2 (Brasseur et al., 2005) |
| Mercator-V1 | NEMO 3.1 (Madec, 2008) | k-esspilon (Gaspar et al., 1990) | Bilaplacian | Isopycnal laplacian | SAM2 + 3DVAR (Lellouche et al., 2013) |
| HCMR | Princeton Ocean Model (Mellor and Blumberg, 1985) | k-l (Mellor and Yamada, 1982) | Laplacian | Bilaplacian | Seek filter (Pham et al., 1998) |
| INGV MP 1 | NEMO 2.3 (Madec, 2008) | (Pacanowski and Philander, 1981) | Bilaplacian | Bilaplacian | no |
| INGV MP 2 | NEMO 2.3 (Madec, 2008) | k-esspilon (Gaspar et al., 1990) | Bilaplacian | Bilaplacian | no |
| INGV MP 3 | NEMO 2.3 (Madec, 2008) | (Pacanowski and Philander, 1981) | Laplacian | Bilaplacian | no |
| INGV MP 4 | NEMO 2.3 (Madec, 2008) | (Pacanowski and Philander, 1981) | Laplacian | Laplacian | no |

**Table 1.** Multi-Physisc Multi-Model SE members model and data assimilation characteristics: the column list the most significant differences between the models in term of code and model physical parameterizations





| MMSE member | STD/STD(OI-SST) | RMSE | ACC |
|---|---|---|---|
| OI-SST | 1.00 | 0.00 | 1.00 |
| INGV-SYS3a2 | 1.16 | 0.16 | 0.91 |
| INGV-SYS4a3 | 1.08 | 0.15 | 0.91 |
| Mercator-V0 | 0.96 | 0.17 | 0.87 |
| Mercator-V1 | 0.81 | 0.19 | 0.81 |
| HCMR | 0.81 | 0.20 | 0.80 |
| INGV MP 1 | 0.82 | 0.17 | 0.87 |
| INGV MP 2 | 0.83 | 0.16 | 0.87 |
| INGV MP 3 | 0.86 | 0.17 | 0.86 |
| INGV MP 4 | 0.87 | 0.17 | 0.86 |

**Table 2.** Dataset statistics, from left to right: Standard Deviation (STD) (with monthly mean seasonal signal removed) normalized by STD of OI-SST , Centered Root Mean Square Error (RMSE) between Members and OI-SST and Anomaly Correlation Coefficient (ACC). All the values were evaluated over the year 2008

| MMSE methods | Extended Name | Main Characteristics |
|---|---|---|
| S1 | Classical SE | Krishnamurti et al. (1999) regression |
| S2 | Classical SE Modified | Seasonal signal removal before the regression |
| S3 | Spatially filtered SE | Seasonal signal removal before the regression and spatial filter applied to the regression estimate |
| S4 | EOFs based SE | EOFs evaluated from observations and members and regression performed on the EOF coefficients. |

**Table 3.** Nomenclature and characteristics of the four MMSE algorithms used





| MMSE data set A (Overconfident Dataset) | MMSE data set B (Well dispersed Dataset) | MMSE data set C (Bad Dispersed Dataset) |
|---|---|---|
| INGV-SYS3a2 | INGV-SYS3a2 | Mercator V1 |
| INGV-SYS4a3 | HCMR | HCMR |
| Mercator V0 | INGV MP1 | INGV MP1 |
| Mercator V1 | INGV MP2 | INGV MP2 |
| HCMR | INGV MP3 | INGV MP3 |
| INGV MP1 | INGV MP4 | INGV MP4 |

**Table 4.** MMSE data sets : members are detailed in Table 1

| SE members | Training periods | | | | | |
|---|---|---|---|---|---|---|
| | 15 days | | 25 days | | 35 days | |
| | S1 | S2 | S1 | S2 | S1 | S2 |
| Full | 1.99 | 1.59 | 1.07 | 0.86 | 0.79 | 0.68 |
| dataset A | 1.19 | 0.96 | 0.77 | 0.65 | 0.62 | 0.56 |
| dataset B | 1.21 | 0.98 | 0.81 | 0.67 | 0.64 | 0.57 |
| dataset C | 1.30 | 1.05 | 0.86 | 0.71 | 0.68 | 0.59 |

**Table 5.** RMSE mean value throughout the Analysis Period for the full dataset (see Table 1 ) and the three datasets of Table 4 as a function of the training period length and the S1 and S2

| Training Period Length(Days) | S1 | S2 | S3 | S4 | BEM |
|---|---|---|---|---|---|
| 15 | 0.35 | 0.56 | 0.89 | 0.77 | 0.91 |
| 25 | 0.62 | 0.72 | 0.86 | 0.78 | 0.91 |
| 35 | 0.68 | 0.75 | 0.84 | 0.77 | 0.91 |

**Table 6.** ACC mean value throughout the Analysis Period for the dataset A (see Table 1) as a function of the training period length for the proposed SE methodologies





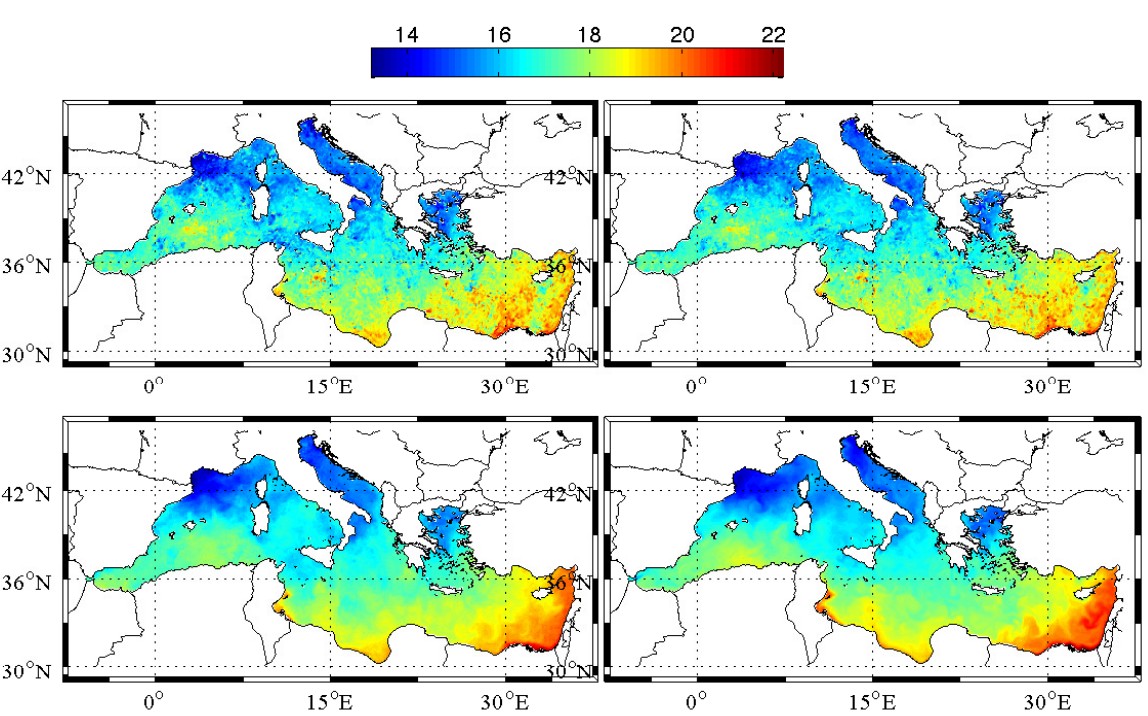

**Figure 1.** MMSE estimates for the first day of the test period (25th of April 2008) using a training period of 15 days, S1 (top panel, left) and the corresponding estimate for S2 (top panel, right), SST from satellite (bottom panel , left ) and Best Ensemble member (bottom panel, right)





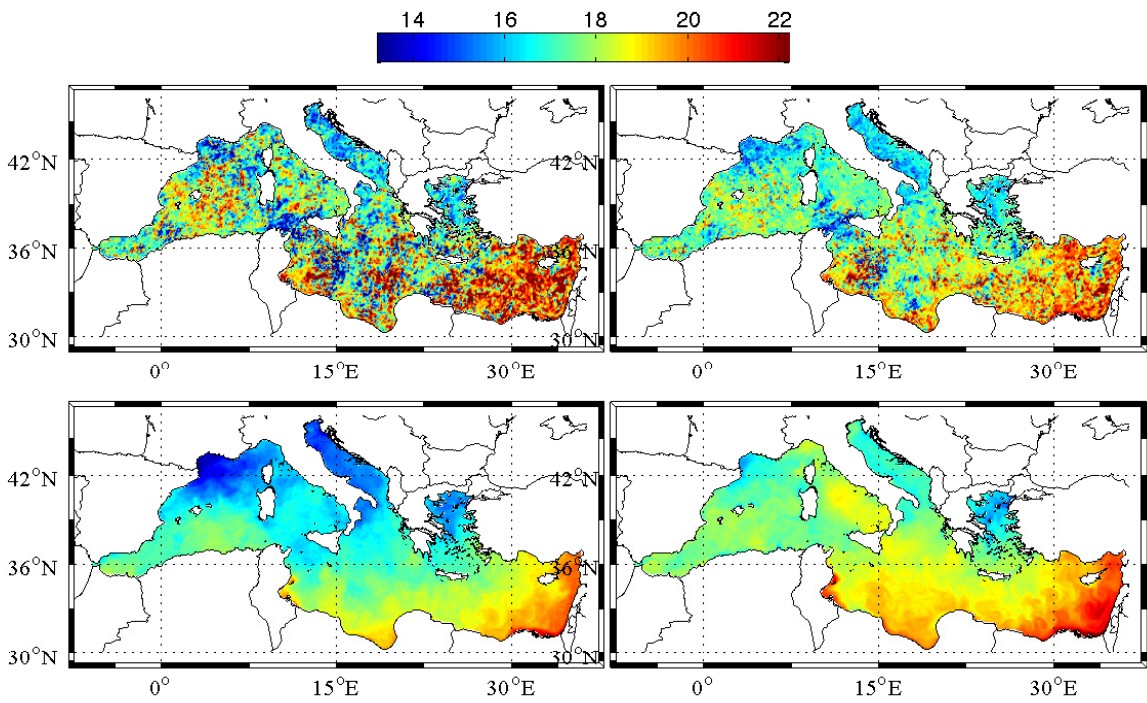

**Figure 2.** MMSE estimates for the last day of test period , the 4th of May 2008, S1 (top panel, left) and the corresponding estimate for S2 (top panel, right), SST from satellite (bottom panel , left ) and Best Ensemble member (bottom panel, right)

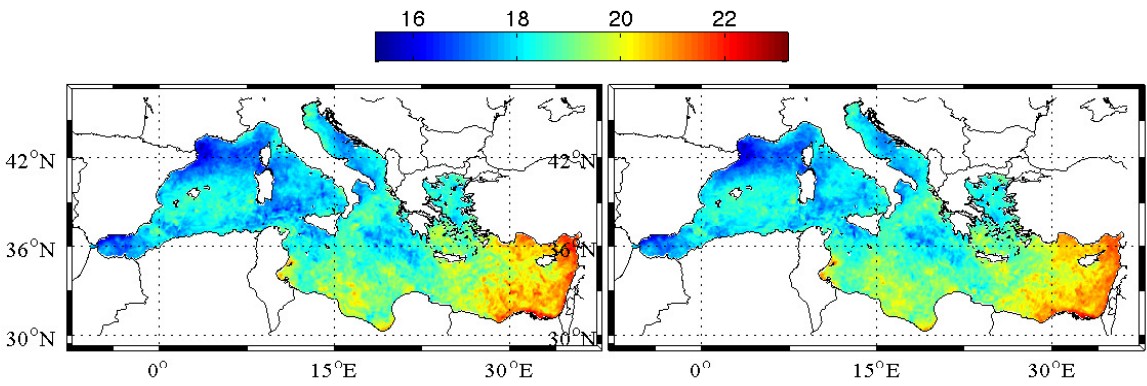

**Figure 3.** MMSE estimates for the first day of the test period (25th of April 2008) using a training period of 35 days, S1 (left panel) and the corresponding estimate for S2 (right panel)





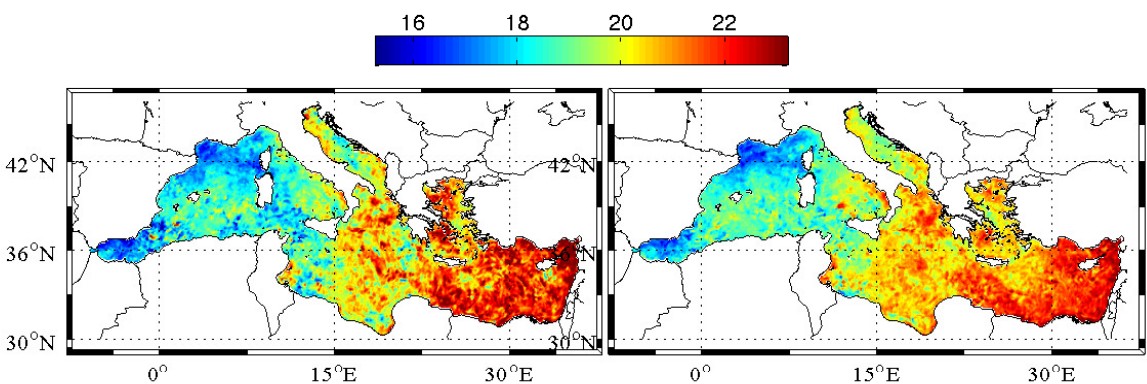

**Figure 4.** MMSE estimates with 35 days training period and for the last day of test period (4th of May2008) S1 (left panel) and the corresponding estimate for S2 (right panel).

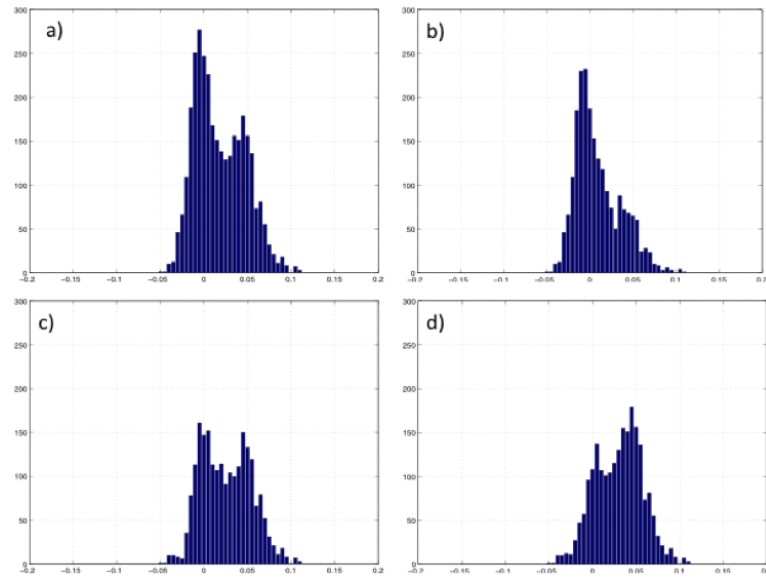

**Figure 5.** Distributions of d in equation 11 for the full dataset (panel a), overconfident dataset B (panel b), well dispersed dataset C (panel c) and badly dispersed dataset D (panel d). The bin-width is $0.05^{o}$ C. Area under the curve equals the total number of models per day in the year 2008.




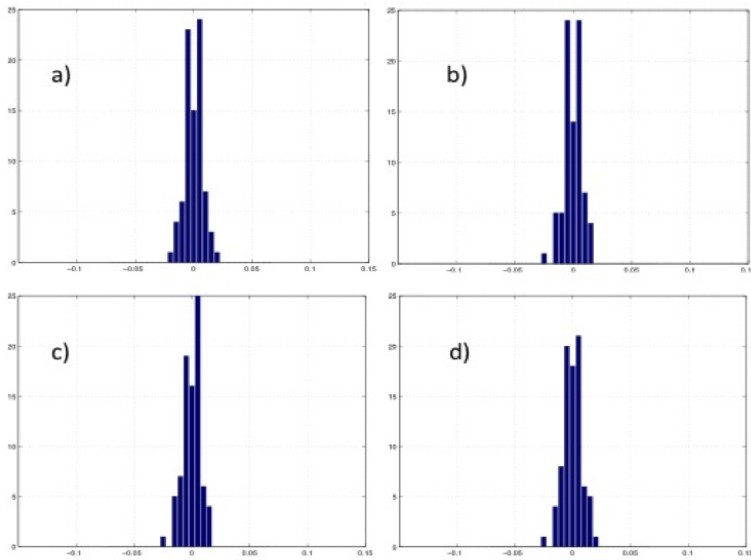

**Figure 6.** the effect of multi-model composition in the distributions of d for the full dataset (panel a), overconfident dataset B (panel b), well dispersed dataset C (panel c) and badly dispersed dataset D(panel d). The bin-width is $0.05^oC$.The effect of multi-model combination of proposed subsample on the SE estimates valid for 1st day of test period with a training period of 14 days

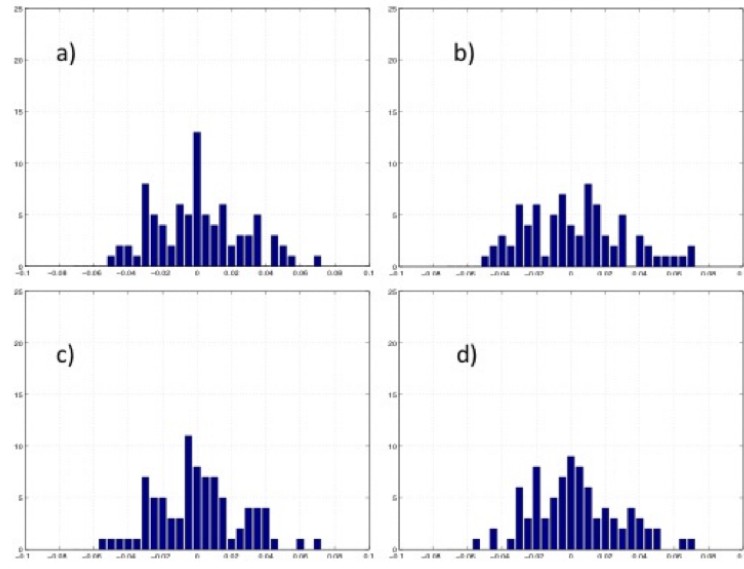

**Figure 7.** the effect of multi-model composition in the distributions of d for the full dataset (panel a), overconfident dataset B (panel b), well dispersed dataset C (panel c) and badly dispersed dataset D(panel d). The bin-width is $0.05^oC$.The effect of multi-model combination of proposed subsample on the SE estimates valid for 10th(last) day of test period with a training period of 14 days





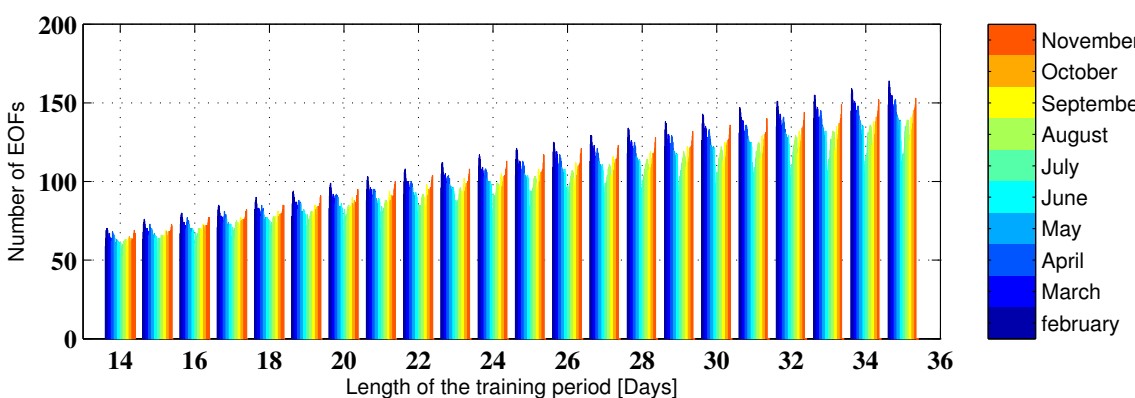

**Figure 8.** Number of retained EOFs histogram, on ordinates the length of the training period, color bar proportioned to the day of the experiments





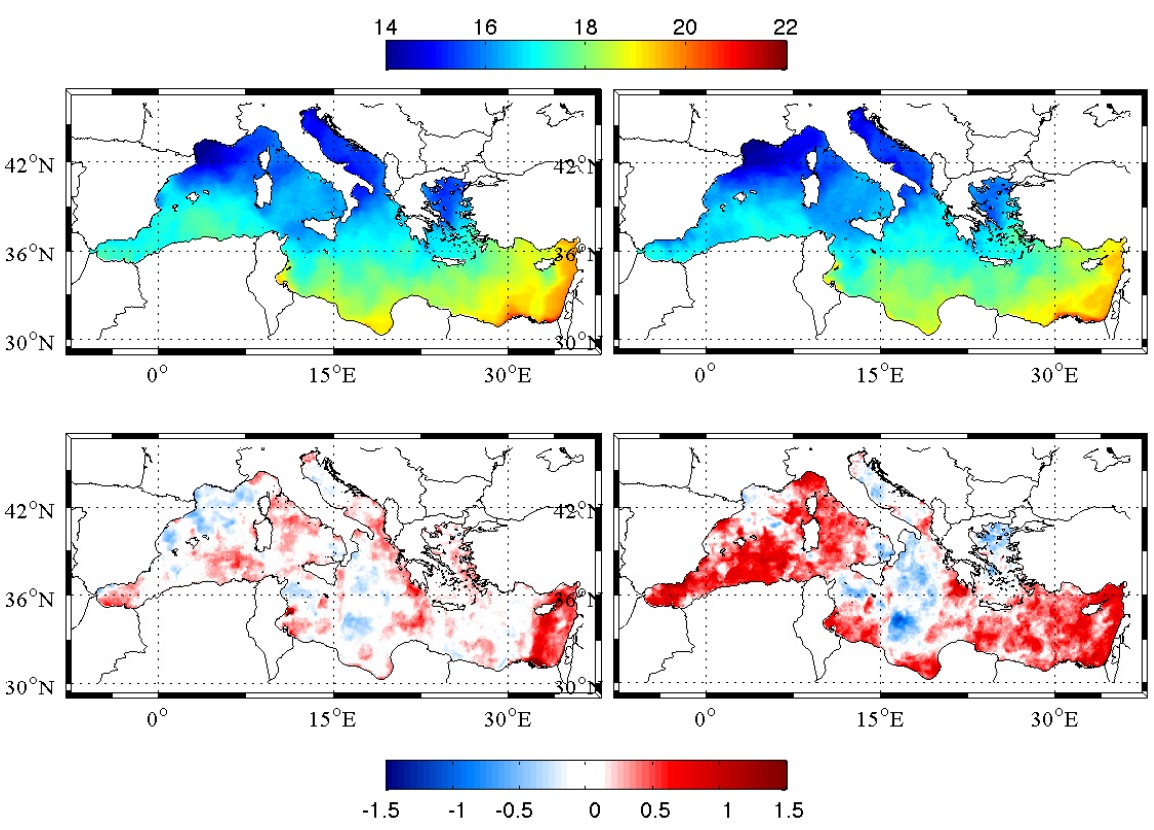

**Figure 9.** S3 and S4 estimate for the first day of the test period (25th of April 2008) using a training period of 15 days, SE3 (top panel, left) and the corresponding estimate for S4 (top panel, right), difference between SST from satellite and S3(bottom panel, left) and difference between SST from satellite and S4(bottom panel, right)





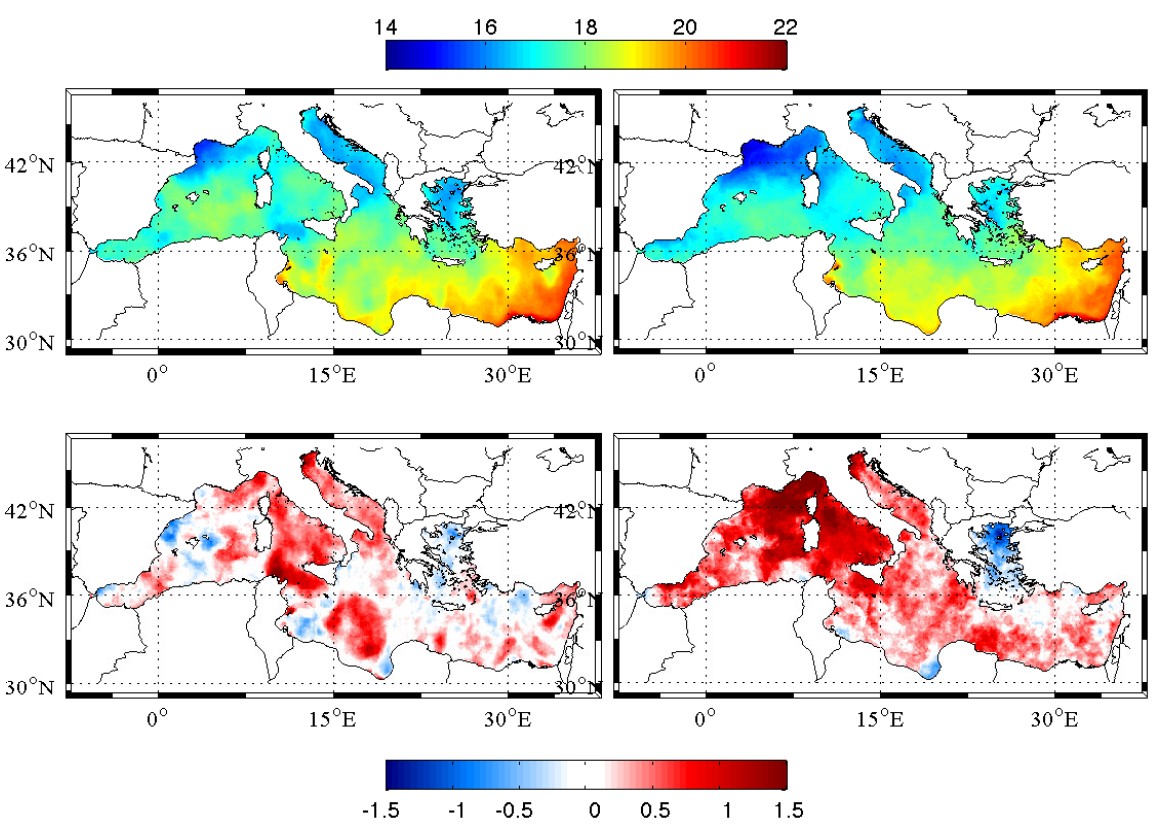

**Figure 10.** S3 and S4 estimates valid the last day of the test period (4th May 2008) using a training period of 15 days, S3 (top panel, left) and the corresponding estimate for S4 (top panel, right), difference between SST from satellite and S3(bottom panel, left) and difference between SST from satellite and S4(bottom panel, right)

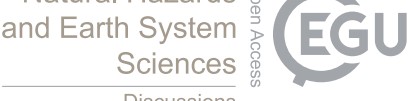



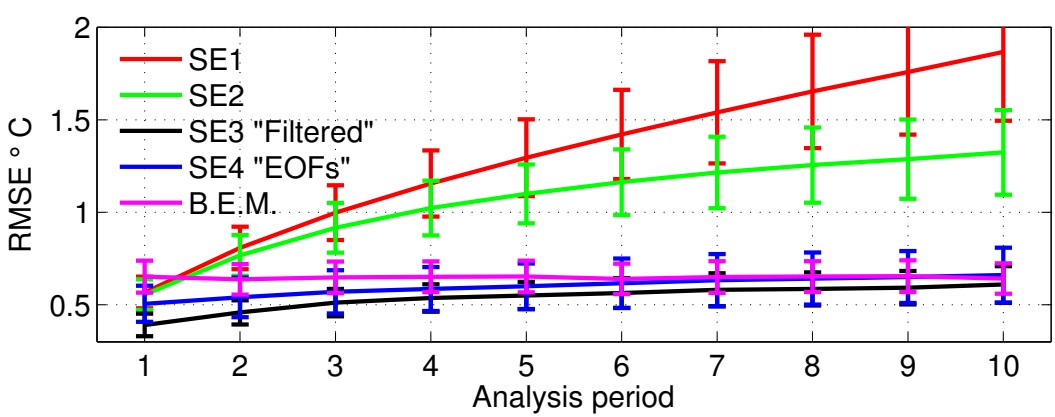

**Figure 11.** Domain average (over the Mediterranean) and time mean over year 2008 of the RMSE for 15 days training period for the overconfident dataset B . Error bar stands for the standard deviation of the RMSE during the year.