# Peer review of "Development of Super-Ensemble techniques for ocean analyses: the Mediterranean Sea case"

_Natural Hazards and Earth System Sciences, 2016_

## Referee Comment (RC1) · Anonymous Referee #1 · 23 May 2016

Overall assessment

The paper entitled "Development of Super-Ensemble techniques for ocean analyses: the Mediterranean Sea case" by J. Pistoia et al describes and compares several regression/super-ensemble techniques applied to ocean analyses in a Mediterranean case study.

Whilst the paper fits well within the scope of the journal, the claimed novel technique using EOFs has already been developed and applied in many different contexts (e.g. Shin and Krishnamurti et al 2003; Rixen et al – several papers; Vandenbulcke et al 2009).

[Figure]

The paper would have offered greater prospects if applied to forecast data as is usually the case with super-ensemble techniques. Applying these methods to analyses raises also a major concern as to the independence of OI-SST reference data, as they may be assimilated in some of the ocean analyses. This may impact the overall interpretation and use of the conclusions. This should be discussed in detail in the paper. Another potential issue is the analyses production cycles which may not be synchronous and hence cause some aliasing in the regression.

The paper should investigate more in depth the sensitivity of the technique to learning periods, as results suggest that the learning might not have 'satured' at 15 or even 35 days. This should be done in conjunction with the selection of the number of modes, which are chosen a priori but could be cross-validated together with the optimization of the learning period instead of picking subsets of models. Likewise, there is no justification for the selection of the 12km filtering radius. There is hence a risk that the sensitivity studies presented in this paper, which do not cover the whole matrix of possible mix of parameters actually miss the optimal combination.

One would wonder if the proposed combination of models also offers interesting 'skill' or properties below the sea surface.

Specific comments

- I would recommend presenting all methods together as anomalies to the reference - it would be interesting to look at the values of weights and see if there is any pattern emerging (model or regional specific for example) - references: add Mourre et al Ocean Dynamics, 2011; Lenartz et al , Ocean Sciences 2010; (and I believe several others from Rixen et al), etc

Other:

- page 1: add space before countries in author list - page 1, line 17, change parenthesis for reference to Pinardi et al - page 7, line 4: comma - page 7, line 22: 'Physisc' - page

8, line 17: 'it's pretty closer'? - page 9, line 2: 'Let us consider to be dialy...'? - page 10, line 2: authors' initials - page 13, caption 'Physisc', 'the column listS' - fig 11: a similar plot should be produced for the anomaly correlation and bias

---

## Referee Comment (RC2) · M. Bostenaru-Dan (Referee) · 5 Jun 2016

The paper presents a topic relevant to the journal. Like the other reviewer, the relationship to other models is not presented clearly enough. The model of Krishnamurty is the outgoing point, but it is not clear if the subsequent equations for regression and calibrations are developed on their own, derived from the other models or part of it. For each equation the source should be named, even if own. Also, I recommend a flow chart for the methodology developed, since this is the main point of the paper. Which models are employed in which point of the methodology. First present the method and then talk about data to train it and the results. This might slightly change the sequencing of the paper. Looking forward for an improved version of the paper.

---

## Author Comment (AC1) · 1 Jul 2016

**Response to the Reviewer's Comments**

Jenny Pistoia, Nadia Pinardi

Authors would like to thank the reviewers for their comments(capital), suggestions and corrections. We have considered all points in our review and answers (in italic).

REVIEWER 1 "OVERALL ASSESSMENT THE PAPER ENTITLED "DEVELOPMENT OF SUPER-ENSEMBLE TECHNIQUES FOR OCEAN ANALYSES: THE MEDITER-RANEAN SEA CASE" BY J. PISTOIA ET AL DESCRIBES AND COMPARES SEVERAL REGRESSION/SUPER-ENSEMBLE TECHNIQUES APPLIED TO OCEAN ANALYSES IN A MEDITERRANEAN CASE STUDY. WHILST THE PAPER FITS WELL WITHIN THE SCOPE OF THE JOURNAL, THE CLAIMED NOVEL TECHNIQUE USING EOFS HAS ALREADY BEEN DEVELOPED AND APPLIED IN MANY DIFFERENT CONTEXTS (E.G.SHIN AND KRISHNA-MURTI ET AL 2003; RIXEN ET AL – SEVERAL PAPERS; VANDENBULCKE ET AL 2009). THE PAPER WOULD HAVE OFFERED GREATER PROSPECTS IF APPLIED TO FORECAST DATA AS IS USUALLY THE CASE WITH SUPER-ENSEMBLE TECHNIQUES"

*The authors thank the reviewer, since in this way we can enhance the innovation proposed in this paper. We will add a section called "Methods used in the literature" in order to discuss the state of the art of the current SE techniques. The basic idea discussed in Krishnamurti's work is that each model can carry somewhat different representation of the foreseen processes, so an appropriate combination can reduce biases in space and time. Here an unbiased linear combination of the available models, optimal (in the least-squares sense) with respect to observations during a training period*

*of a priori chosen length reduces the RMSE both for prediction on the south-north component of winds at 850 hPa averaged over bounded between 50°E and 120°E , Asian monsoon Monsoon precipitation simulations and Hurricane track and intensity forecasts. Since all observations have equal importance. Lenartz et al. (2010) applied this method for ocean wave forecasting introducing a way to change the importance in the observation using data assimilation techniques (Kalman filter and particle filter) adapted to the super-ensemble paradigm. With this technique the regression weights change on a time-scale corresponding to their natural characteristic time, discarding older information automatically and rate of change is determined by the uncertainties of the weights, models and of observations. Rixen at al, 2009 in very limited area could demonstrate that the SE methods outperforms the individual models on several error measures. Skill improvements can be found applying dynamic, non-Gaussian and regularized filters. Our study significantly differs from the previous methods because for the first time we assess the SE prediction impact in term of the dataset composition. We begin our research activity studying the characteristic that a dataset should fulfil in term of the spread of the ensemble and the mean bias of each member. Thus only a multi-model multi-physics dataset could satisfy all the requirements.*

*A proof is a second set of experiments carried out with two subsamples:*

- *Subsample D: Multi Model MM: INGV SYS3A2 INGV sys4a3 Mercator V0, Mercator V1;HCMR*

- *Subsample E: Multi Model Multi Physics: MM-MP INGV SYS3A2 INGV sys4a3 Mercator V1;HCMR and INGV MP1*

*It is clear from figure 1 below the improvements bring on the SE substituting one member with a simulation with similar performances.*

"APPLYING THESE METHODS TO ANALYSES RAISES ALSO A MAJOR CONCERN AS TO THE INDEPENDENCE OF OI-SST REFERENCE DATA, AS THEY MAY BE ASSIMILATED IN

SOME OF THE OCEAN ANALYSES. THIS MAY IMPACT THE OVERALL INTERPRETATION AND USE OF THE CONCLUSIONS. THIS SHOULD BE DISCUSSED IN DETAIL IN THE PAPER"

*The authors thank the reviewer for stressing the point. We decided to use the OISST because we would like applying the SE paradigm to a regional ocean model. As discussed in "Methods used in the literature" SE in ocean has been applied to limited area of the Mediterranean Sea (Ligurian Sea and small portion of Adriatic). We need of a robust and reliable truth estimator valid over the whole Mediterranean, since SE method can be successfully applied on observations that can cover the same degree of freedom of dataset. Satellites data as SST or SSH can be optimally interpolated to get 2D maps. The alternative of remote sensing information would be to compare models with in-situ observations but they are too sparse in space and time. For example only a dozen of ARGO floats drift in the sea and can't be used. The mooring buoy aren't reliable in time, moreover we neglect two factors: buoy position near coast where the coastline is not well resolved in the grid models and the representativeness error of the observation compared with grid point model value. However the authors would agree with reviewers if the OI-SST is assimilated in the MM-MP members, but none of the model employed has assimilated the OI-SST. There is only flux relaxation through SST nudging for INGV set up and Mercator. Moreover HCMR and Mercator assimilate the GOS AVHRR SST.*

*"Another potential issue is the analyses production cycles which may not be synchronous and hence cause some aliasing in the regression." We think that the unbiasing procedure together with the seasonal signal removed to all models and observations prior the regression should limit any aliasing in the procedure.*

"THE PAPER SHOULD INVESTIGATE MORE IN DEPTH THE SENSITIVITY OF THE TECHNIQUE TO LEARNING PERIODS, AS RESULTS SUGGEST THAT THE LEARNING MIGHT NOT HAVE 'SATURED' AT 15 OR EVEN 35 DAYS. THIS SHOULD BE DONE IN CONJUNCTION WITH THE SELECTION OF THE NUMBER OF MODELS, WHICH ARE CHOSEN A PRIORI BUT

COULD BE CROSS-VALIDATED TOGETHER WITH THE OPTIMIZATION OF THE LEARNING PERIOD INSTEAD OF PICKING SUBSETS OF MODELS."

*In order to In order to find the minimum training period length possible a simple experiment has been done using the observations as one of the ensemble members in the training period. This test can be considered as the maximum skill that could be achieved with a MMSE approach, and it is also a way to check the coefficient estimates. For a training period of 15 days, all the regression coefficients are zero, except the weight related to the observational member, which is retrieved to be 1. Trimming the dataset (removing members) we noticed that when the training period days (M) are less than the number of ensemble members involved (N), in our case 9 (Table 1), the algorithm fails, giving incorrect values for the coefficients. Hence the minimum training period units must be such that M > N. In our case any training period longer than 10 days will work well. However to add robustness to regression algorithm we set 15 days as the minimum length of learning period. Even if we would neglect the overfitting that affect SE prediction trained with learning period longer 35 days, we think long training are out of the scope of our research since we are focused on a potential operative approach.*

"LIKEWISE, THERE IS NO JUSTIFICATION FOR THE SELECTION OF THE 12KM FILTERING RADIUS. THERE IS HENCE A RISK THAT THE SENSITIVITY STUDIES PRESENTED IN THIS PAPER, WHICH DO NOT COVER THE WHOLE MATRIX OF POSSIBLE MIX OF PARAMETERS ACTUALLY MISS THE OPTIMAL COMBINATION."

The right radius value is 15km (and not 12kn as erroneous written). This value has been found by means of sensitivity studies done applying a circular filter in each point of the domain. Below figure shows RMSE according the chosen filter radius length. We see that with short radius there is no influence of the filtering. With radius longer 15km the fields became too smooth and there is degradation. Thus best performances are reached with a filter of 15km radius.

"ONE WOULD WONDER IF THE PROPOSED COMBINATION OF MODELS ALSO OFFERS INTERESTING 'SKILL' OR PROPERTIES BELOW THE SEA SURFACE."

*Observations are scarce below the surface and thus it is very difficult to envisage a horizontal EOF method that takes advantage from the observation itself. Moreover it is difficult to propagate the information of SE in sub-surface avoiding model shocks.*

SPECIFIC COMMENTS - I WOULD RECOMMEND PRESENTING ALL METHODS TOGETHER AS ANOMALIES TO THE REFERENCE - IT WOULD BE INTERESTING TO LOOK AT THE VALUES OF WEIGHTS AND SEE IF THERE IS ANY PATTERN EMERGING (MODEL OR REGIONAL SPECIFIC FOR EXAMPLE) - REFERENCES: ADD MOURRE ET AL OCEAN DYNAMICS, 2011; LENARTZ ET AL , OCEAN SCIENCES 2010; (AND I BELIEVE SEVERAL OTHERS FROM RIXEN ET AL), ETC

*We have done some figures for the weights distribution but no pattern appear. In general weights are bigger for the S1 and S2 (figure 3) compared with S4 calculation(figure 4). It is clear that the regression need time to evaluate the weights and for this reason the weights obtained with short trained period are noisy field, and this characteristic is depicted in the S1 and S2 maps.*
*We follow your advice and we replaced figures in the paper as OI-SST anomaly*

OTHER: - PAGE 1: ADD SPACE BEFORE COUNTRIES IN AUTHOR LIST - PAGE 1, LINE 17, CHANGE PARENTHESIS FOR REFERENCE TO PINARDI ET AL - PAGE 7, LINE 4: COMMA - PAGE 7, LINE 22: 'PHYSICS' - PAGELINE 17: 'IT'S PRETTY CLOSER'? - PAGE 9, LINE 2: 'LET US CONSIDER TO BE DIALY: : :'? — PAGE 10, LINE 2: AUTHORS' INITIALS - PAGE 13, CAPTION 'PHYSICS', 'THE COLUMN LISTS' - FIG 11: A FIXED SIMILAR PLOT SHOULD BE PRODUCED FOR THE ANOMALY CORRELATION AND BIAS DONE

---

## Author Comment (AC3) · 1 Jul 2016

July 1, 2016

Reviewer 2

THE PAPER PRESENTS A TOPIC RELEVANT TO THE JOURNAL. LIKE THE OTHER RE-VIEWER, THE RELATIONSHIP TO OTHER MODELS IS NOT PRESENTED CLEARLY ENOUGH. THE MODEL OF KRISHNAMURTY IS THE OUTGOING POINT, BUT IT IS NOT CLEAR IF THE SUBSEQUENT EQUATIONS FOR REGRESSION AND CALIBRATIONS ARE DEVELOPED ON THEIR OWN, DERIVED FROM THE OTHER MODELS OR PART OF IT. FOR EACH EQUATION THE SOURCE SHOULD BE NAMED, EVEN IF OWN.

ALSO, I RECOMMEND A FLOW CHART FOR THE METHODOLOGY DEVELOPED, SINCE THIS IS THE MAIN POINT OF THE PAPER. WHICH MODELS ARE EMPLOYED IN WHICH POINT OF THE METHODOLOGY. FIRST PRESENT THE METHOD AND THEN TALK ABOUT DATA TO TRAIN IT AND THE RESULTS. THIS MIGHT SLIGHTLY CHANGE THE SEQUENCING OF THE PAPER. LOOKING FORWARD FOR AN IMPROVED VERSION OF THE PAPER.

*The authors thank the reviewer for this comment. Equations have been checked. Equations from 1 to eq. 4 come from Krishnamurti's work. The others are a manipulation of the initial system. However a flow chart has been added to help the reader.*

**Fig. 1.** Paper Flowchart